# Experimental and Simulation-Based Performance Analysis of a Computed Torque Control (CTC) Method Running on a Double Rotor Aeromechanical Testbed

**Árpád Varga** [1] , **György Eigner** [2,*], **Imre Rudas** [3] and **József Kázmér Tar** [1,3]

1   Doctoral School of Applied Informatics and Applied Mathematics, Óbuda University,
    H-1034 Budapest, Hungary; varga.arpad@uni-obuda.hu (Á.V.); tar.jozsef@nik.uni-obuda.hu (J.K.T.)
2   Physiological Controls Research Center, Óbuda University, H-1034 Budapest, Hungary
3   Antal Bejczy Center for Intelligent Robotics, Óbuda University, H-1034 Budapest, Hungary;
    rudas@uni-obuda.hu
*   Correspondence: eigner.gyorgy@nik.uni-obuda.hu

**Abstract:** Concept of closed loop control appears in many fields of engineering sciences, where the output quantity of some physical system must be forced to follow some prescribed function over time, e.g., when a robotic arm endpoint must track a desired trajectory or path given as timed series of spatial coordinates. The classic approach for solving this kind of problem involves a PID compensation block, and the necessary input signal for keeping the controlled process in the vicinity of the desired trajectory is calculated as the weighted sum of momentary deviation, deviation integral, and deviation derivative relative to the reference path. However, despite the obvious advantages, practical usability, and simplicity of the PID controllers, their performance is limited when they are utilized for controlling nonlinear systems. Even with linear systems, their proper operation requires an accurate system model and precise tuning process for finding the best weight values for the proportional, integral, and derivative effects, and the planned closed loop behavior might change significantly as the parameters of the controlled plant change over time. In this article, a computed torque-based controller is presented, which has only one adjustable parameter ensuring precise trajectory tracking even with significantly alternated model constants. The practical usability of the offered algorithm is evaluated and verified by simulations and experiments performed on a simple mechanical bi-rotor testbed playing the role of controlled plant.

**Keywords:** Computed Torque Control; robust control; model based control; experiments on control performance; control simulations

## 1. Introduction

The traditional formulation of the Computed Torque Control (CTC) assumes the possession of a precise system model and the lack of unknown or unobserved external disturbances, known nominal trajectory to be tracked $q^N(t)$, the actual trajectory $q(t)$ according to (1) in which $Q(t)$ denotes the generalized forces exerted by the robot drives, $H(q)$ being a positive definite (sometimes not very well conditioned but at least in principle invertible) inertia matrix of the robot arm, and $h(q, \dot{q})$ containing Coriolis and gravitational forces:

$$e(t) := e^N(t) - q(t) \ , \ e_{int}(t) = \int_{t_0}^{t} e(\xi) \mathrm{d}\xi \ , \tag{1a}$$

$$H(q)\ddot{q} + h(q, \dot{q}) = Q \ , \tag{1b}$$

$$H(q)\Big[q^N(t) + K_I e_{int}(t) + K_P e(t) + K_D \dot{e}(t)\Big] + h(q, \dot{q}) = Q \ . \tag{1c}$$

It is assumed that $q(t)$ and $\dot{q}(t)$ can be measured by industrial sensors. By subtracting (1b) from (1c) and utilizing the existence of $H(q)^{-1}$, it is obtained that

$$\ddot{e}(t) + K_I e_{int}(t) + K_P e(t) + K_D \dot{e}(t) \equiv 0 \ . \tag{2}$$

Accordingly, the integral ($K_I$), the proportional ($K_P$), and the derivative ($K_D$) feedback gains must be appropriately set to guarantee the $e_{int}(t) \to 0$, $e(t) \to 0$, and $\dot{e}(t) \to 0$ as $t \to \infty$. The traditional way is based on the introduction of a "artificial state variable" $x(t) = [e_{int}(t), e(t), \dot{e}(t)]^T$ that evidently satisfies the LTI systems' equation of motion in (3).

$$\dot{x} = \begin{bmatrix} e(t) \\ \dot{e}(t) \\ \ddot{e}(t) \end{bmatrix} = \begin{bmatrix} 0 & I & 0 \\ 0 & 0 & I \\ -K_I & -K_p & -K_D \end{bmatrix} \begin{bmatrix} e_{int}(t) \\ e(t) \\ \dot{e}(t) \end{bmatrix} \equiv Ax \ . \tag{3}$$

By investigating the Jordan canonical form of matrix $A$ (e.g., [1]), it can be stated that the system is stable if and only if the real part of each eigenvalue of matrix $A$ in (3) is negative.

A more traditional approach is considering the symmetric positive definite matrix $Q$ and the matrix function $\Phi \in \mathbb{R}^{n \times n}(\xi)$ defined as

$$\Phi(\xi) \stackrel{def}{=} e^{\xi A^T} Q e^{\xi A}, \ \xi \in \mathbb{R} \ . \tag{4}$$

Evidently, $\Phi(\xi)$ is symmetric, positive definite since the inverse of $e^{\xi A^T}$ is $e^{-\xi A^T}$ and the inverse of $e^{\xi A}$ is $e^{-\xi A}$, i.e., these matrix exponential functions are invertible, therefore, they cannot map a nonzero array to zero. The time-derivative of $\Phi(\xi)$ can be computed as

$$\frac{\mathrm{d}}{\mathrm{d}\xi}\Phi(\xi) = A^T \Phi(\xi) + \Phi(\xi) A \text{ from which it follows that} \tag{5a}$$

$$\Phi(t_1) - \Phi(t_0) = \int_{t_0}^{t_1} \frac{\mathrm{d}}{\mathrm{d}\xi}\Phi(\xi)\mathrm{d}\xi \equiv A^T \Psi(t_0, t_1) + \Psi(t_0, t_1) A \ , \tag{5b}$$

in which

$$\Psi(t_0, t_1) \stackrel{def}{=} \int_{t_0}^{t_1} \Phi(\xi)\mathrm{d}\xi \ . \tag{6}$$

In connection with the Canonical Form of Quadratic Matrices by Jordan [1,2], $\Psi(\infty)$ exists only if each eigenvalue of the matrix $A$ has a negative real part. In this case $e^{\xi A} \to 0$ as $\xi \to \infty$, $\Phi(\infty) = 0$, and $\Psi(t_0, \infty)$ is finite. For the special case of $t_0 = 0$, (5) yields that

$$-Q = A^T \Psi(0, \infty) + \Psi(0, \infty) A \equiv A^T P + PA \ , \tag{7}$$

in which evidently $P \equiv \Psi(0, \infty) \stackrel{def}{=} \int_0^\infty e^{\xi A^T} Q e^{\xi A} \mathrm{d}\xi > 0$ (i.e., positive definite). For solving the so-called Lyapunov equation defined by (7), modern program languages, such as e.g., Julia language [3], simple functions are available by the use of which it can be checked whether a given setting of the parameters $K_I$, $K_P$, and $K_D$ and given matrix $Q$ results in a solvable Lyapunov equation. This approach has a close relationship with the idea of PID controller.

The CTC controller can be introduced in a little bit different manner outlined in Figure 1.

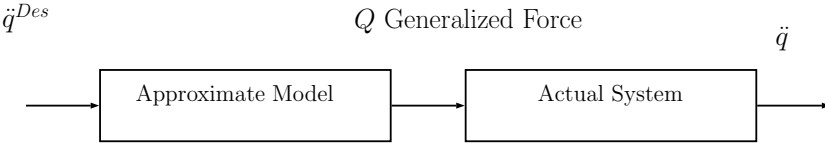

**Figure 1.** Interpretation of the CTC scheme in different manners.

Assume that, for a second order system, on the basis of purely kinematic considerations, a "desired" $\ddot{q}^{Des}(t)$ 2nd time-derivative is constructed that guarantees the $e_{int}(t) \to 0$, $e(t) \to 0$, and $\dot{e}(t) \to 0$ as $t \to \infty$ conditions if it is realized. If the available system model is not exactly precise (i.e., it contains the $\hat{H}(q), \hat{h}(q, \dot{q})$ approximate model) while the exact model contains the functions $H(q), h(q, \dot{q})$, in Figure 1, the control force is computed from the approximate model, and the realized $\ddot{q}(t)$ value will be

$$\ddot{q} = H(q)^{-1}\left[\hat{H}(q)\ddot{q}^{Des} + \hat{h}(q, \dot{q}) - h\right] , \tag{8}$$

that in the special case of the "exact" approximate model leads to $\ddot{q}(t) = \ddot{q}^{Des}(t)$. In this approach for designing $\ddot{q}^{Des}(t)$, we can follow a simpler choice than the more general PID-based approach. Let $0 < \Lambda = $ const. and try to realize the motion according to

$$\left(\Lambda + \frac{\mathrm{d}}{\mathrm{d}t}\right)^3 e_{int}(t) \equiv 0 . \tag{9}$$

Since for differentiable $f(t)$ functions $\Lambda \mathrm{d}f/\mathrm{d}t \equiv \mathrm{d}(\Lambda f)/\mathrm{d}t$, (9) leads to

$$\ddot{q}^{Des}(t) = \ddot{q}^N(t) + \Lambda^3 e_{int}(t) + 3\Lambda^2 e(t) + 3\Lambda\dot{e}(t), \tag{10}$$

which evidently corresponds to particular possible PID feedback terms depending only on a single parameter $\Lambda$. To show that this setting results in vanishing tracking error as $t \to \infty$, consider the functions $g_k(t) := (t - t_0)^k e^{-\Lambda(t-t_0)}$. Since

$$\dot{g}_k(t) = kg_{k-1}(t) - \Lambda g_k(t), \tag{11}$$

it follows that $\left(\Lambda + \frac{\mathrm{d}}{\mathrm{d}t}\right)^i g_{i-1}(t) = 0$. Consequently, the general solution of equations in (9) for $e_{int}(t)$ is

$$e_{int}(t) = \sum_{\ell=0}^{2} c_\ell g_\ell(t) , \tag{12}$$

in which the free coefficients $\{c_0, c_1, c_2\}$ can be chosen according to the initial conditions. In other words, the linear set of the solutions is spanned by the basis vectors $\{g_0(t), g_1(t), g_2(t)\}$ so that each basis vector converges to 0 as $t \to \infty$. It can be noted that, for tackling the problem of modeling errors, the traditional approach, such as the "Adaptive Inverse Dynamics Controller" or the "Adaptive Slotine-Li Controller" [4], evolves by the use of Lyapunov's stability theorem and his 2nd or "direct" method [5,6]. The "Robust Variable Structure / Sliding Mode Controller" invented in the Soviet Union in the past century (e.g., [7–9]), instead of trying to realize (9) (that according to the computations is very sensitive to the modeling errors), introduced the concept of "error metrics" $S(t)$ and the control goal

$$S(t) := \left(\Lambda + \frac{\mathrm{d}}{\mathrm{d}t}\right)^2 e_{int}(t) \tag{13a}$$

$$\dot{S}(t) \approx -K\tanh\left(\frac{S}{w}\right) \tag{13b}$$

with $K > 0$, $w > 0$ parameters. The simple idea behind (13) is that, for big $S$ components, the "tanh" function is saturated at $\pm 1$; therefore, $S$ can be driven to the vicinity of zero during finite time and subsequently can be kept around zero. The subtle dynamic details of how $S(t)$ is driven to zero or how is it kept near zero in this approach are not important; for this reason, a very approximate model can work well to maintain the $S(t) = \left(\Lambda + \frac{\mathrm{d}}{\mathrm{d}t}\right)^2 e_{int}(t) \approx 0$ that has quite similar consequences as (9) for the error integral and the error. This solution evidently can suffer from the phenomenon of chattering if parameter $K$ is very big and the "smoothing parameter" $w$ is too small.

Recently, control algorithms based on or incorporating CTC have been used in a wide variety of applied engineering research areas, such as motion control of miscellaneous robot manipulators with open and closed (or parallel) kinematic chains [10,11] and cable-driven robots [12]; overhead crane payload sway control [13]; attitude control of drone-like multi-rotor aircraft [14]; operation of a musculoskeletal therapy device with artificial muscles [15]; gait planning for bipedal robots [16]. However, in almost all cases, separate integral, proportional, and derivative gains or parameters are used to "tune" the controlled system for the optimal trajectory tracking, which makes finding the ideal parameter set a complicated task.

The methodology followed in the article is as follows: first, a simple real-world testbed for control algorithms is introduced and its mathematical model is constructed based on various parameter estimation measurements (Figure 2). Then, the performance of the one-parameter CTC algorithm is studied by comparing the simulated and measured (when the algorithm was running on a microcontroller, controlling the real-world testbed) trajectory tracking results. Further experiments were carried out on the real-world testbed showing trajectory tracking with alternative trajectory profiles (sine signal with increasing frequency). The resistance against outer disturbances tested with simulation and measurement as well and the results were compared qualitatively. Besides that, an alternative, more traditional control method (PID compensator with nonlinear feedforward term, referred to as "Nonlin. PID" in Figure 2) is also tested for trajectory tracking by simulation and measurement. Finally, the robustness of CTC and "Nonlin. PID" methods against model parameter variations (when the controller, tuned for the original plant model, interacts with a plant with altered parameters) were compared using the simulational results of the two methods.

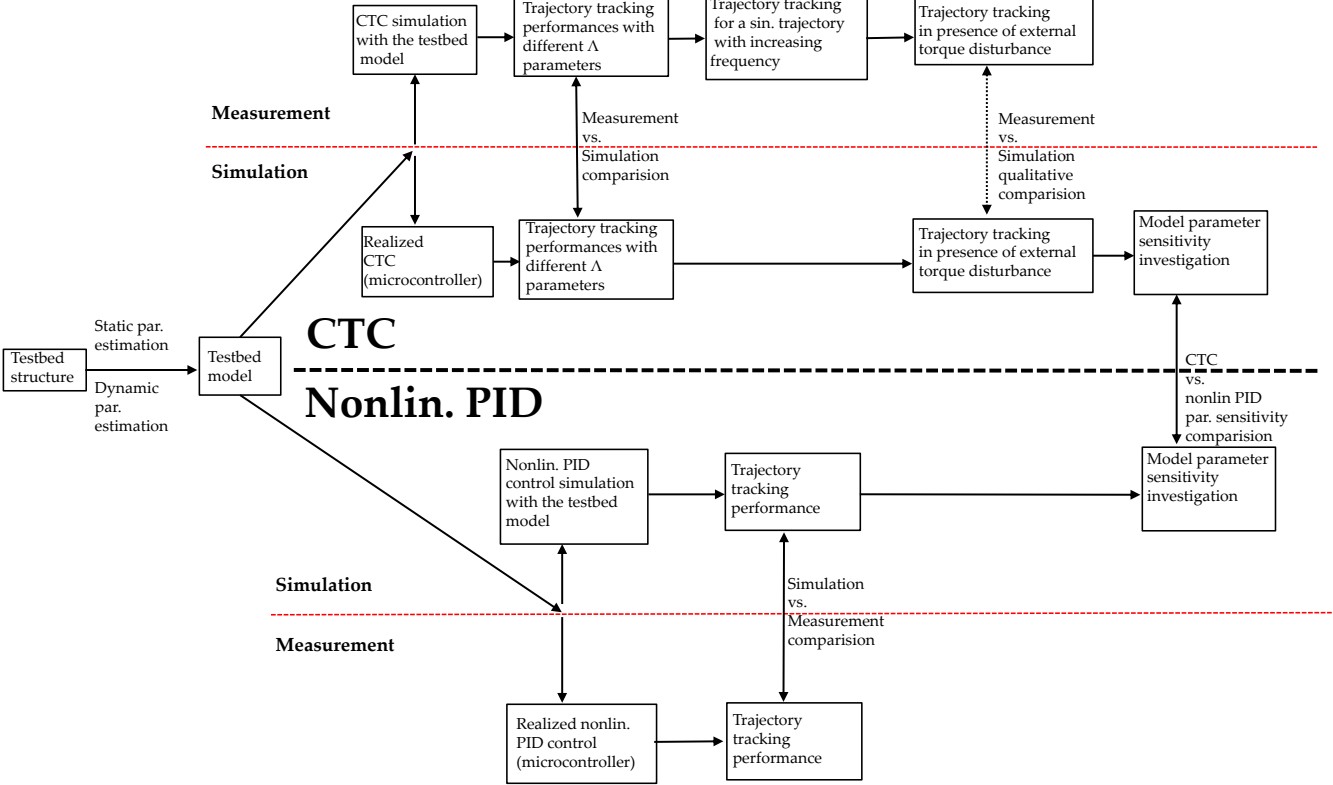

**Figure 2.** Block diagram of the methodology used in the article.

## 2. Structure and Mechanical Model of the Bi-Rotor Testbed

To justify the practical usability of the proposed one parameter CTC scheme, a simple twin-rotor experimental test stand was built, consisting of a beam rotating in the vertical plane around a horizontal axis in the vicinity of its middle point. The angular position of the arm—which is the controlled variable—can be varied by a control torque exerted by two propellers at both ends of the rod. Propellers were driven by electric motors with variable rotational speed, which can be regulated by a Pulse-Width Modulated (PWM) electric signal. The test platform is also equipped with a programmable microcontroller board on which the control algorithm can be implemented and executed in real-time. A photograph of the device and free body diagram of the arm belonging to the testbed is shown in Figure 3. Basic information about the main components of the device is summarized in Table 1.

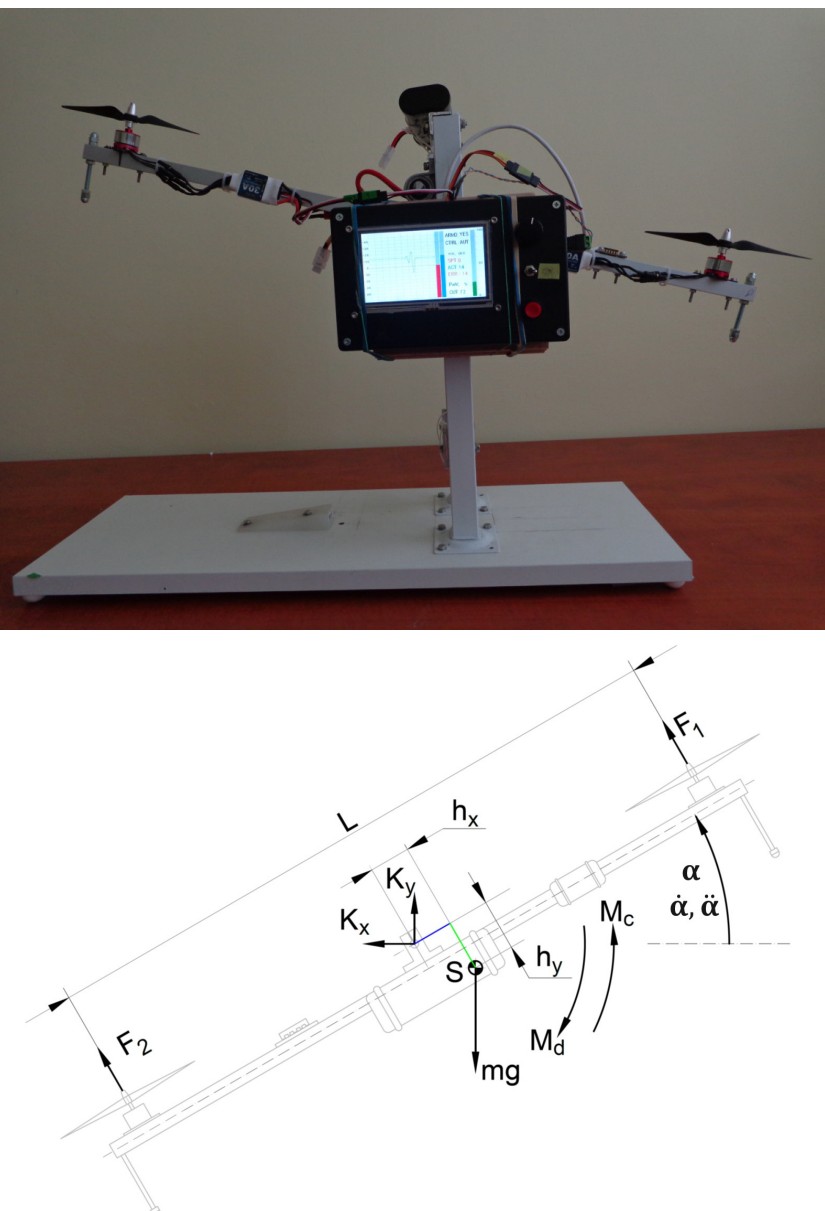

**Figure 3.** (**Above**) The double rotor test stand; (**Below**) Free body diagram of the bar, belonging to the double rotor testbed.

**Table 1.** Information about the main bi-rotor test bed components.

| Parameter Name | Type and/or Manufacturer, Main Characteristics |
|---|---|
| motor (2 pcs) | Brushless Direct Current (BLDC) C20-1550 kv |
| Electric Speed Controller (ESC, 2 pcs) | redox 30 A controlled with PWM signal |
| propeller blade (2 pcs) | ⌀ 150 mm |
| tilt sensor | Murata SCA100T-D02 meas. range: −90 to +90 deg precision: ±0.86 deg analog 0–5 V output used for angle reading |
| controller board | Adafruit Metro M4 micro-controller: Microchip ATSAMD51 Cortex M4 core running at 120 MHz |
| battery | Nickel Metal Hydrid (NIMH) 7.2 V, 3600 mAh, 6 cell rechargable |
| display | Nextion NX8048T050 |

The position of the lever arm is characterized by the inclination angle $\alpha$ given in degrees, which is defined as the angle between the horizontal direction (perpendicular to the gravitational gradient vector g) and the center-line of the arm. The instantaneous value of thrust force vectors generated by the left and right propellers are denoted by $F_1(t)$ and $F_2(t)$. The distance between the center lines of the two rotors is $L$. The gravitational force $mg$ (where $m$ is the mass of the arm) is acting at the $S$ center of gravity. The location of point $S$ relative to the rotational center point is given by distances $h_x$, $h_y$, where $h_x$ is measured parallel with the arm center-line and $h_y$ is measured perpendicular to the arm center-line. The horizontal and vertical components of the reaction force from the bearing are denoted by $K_x$ and $K_y$. $\alpha(t)$, $\dot{\alpha}(t)$ and $\ddot{\alpha}(t)$ are the instantaneous angular position, angular velocity, and angular acceleration of the lever arm, respectively. The damping torque $M_d$ is considered to be proportional to the actual angular velocity:

$$M_d(t) = d\dot{\alpha}(t), \tag{14}$$

where $d$ is a damping constant. By considering the aforementioned forces and torques, the lever arm equation of motion can be written in the following form:

$$\Theta\ddot{\alpha}(t) = M_c(t) - mg[h_x cos\alpha(t) + h_y sin\alpha(t)] - d\dot{\alpha}(t), \tag{15}$$

where $\Theta$ is the lever arm moment of inertia calculated to the rotational center. $M_c(t)$ is the actual control torque, the second term in the equation is the torque of the gravitational force, and the third is the damping torque. Since the reaction forces $K_x$ and $K_y$ are rising at the center of rotation, they do not produce torque contribution. The connection between the $u_1(t)$, $u_2(t)$ input PWM signals (to the left and right motors) and the exerted $F_1(t)$, $F_2(t)$ thrust forces can be modeled as first order linear functions:

$$F_1(t) = Au_1(t), F_2(t) = Au_2(t). \tag{16}$$

This consideration can be only held when $u_1(t)$ or $u_2(t)$ varies very slowly or when they are stationary constant values. Parameter $A$ is already known from experiments with the single-arm testbed. The actuator (BLDC motor with the propeller) dynamics is modeled with a dead-time, first-order block with parameters $\tau$ force delay time and $t_{rise}$

force rising time, which was also measured previously when the single-arm version of the testbed was built (for more details about the force exertion dynamics, see [17]):

$$F_1(t) = A[u_1(t - \tau) - t_{rise}\dot{u}_1(t - \tau)]; F_2(t) = A[u_2(t - \tau) - t_{rise}\dot{u}_2(t - \tau)] \quad (17)$$

The left and right motor PWM control signals are calculated from the single incoming control signal $u(t)$ according to the following logical rules:

$$u_1(t) = u_{lowest} + |u(t)|, \text{ when } u_{max} > u(t) > 0;$$
$$u_1(t) = u_{lowest}, \text{ when } u(t) \leqslant 0$$
$$u_1(t) = u_{highest}, \text{ when } u(t) \geq u_{max}$$
$$u_2(t) = u_{lowest} + |u(t)|, \text{ when } u_{min} < u(t) < 0;$$
$$u_2(t) = u_{lowest}, \text{ when } u(t) \geqslant 0$$
$$u_2(t) = u_{highest}, \text{ when } u(t) \leq u_{mim}.$$

The control signal distribution logic is also illustrated in Figure 4.

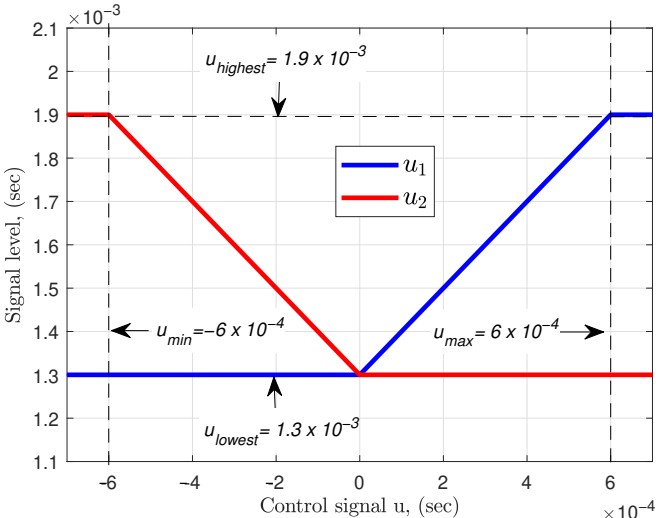

**Figure 4.** Distribution of the input signal $u(t)$ between the two rotors.

Current control torque $M_c(t)$ generated by the two propellers is:

$$M_c(t) = \frac{L}{2}(F_2(t) - F_1(t)). \quad (18)$$

Therefore, momentary control torque acting on the arm written with the input signal $u$ is:

$$M_c(t) = A\frac{L}{2}[u(t - \tau) - t_{rise}\dot{u}(t - \tau)]. \quad (19)$$

The equation of motion, including the actuation dynamics, becomes:

$$\Theta\ddot{\alpha}(t) + d\dot{\alpha}(t) + f(\alpha(t)) = A\frac{L}{2}[u(t - \tau) - t_{rise}\dot{u}(t - \tau)], \quad (20)$$

where $f(\alpha(t))$ is a nonlinear function $-mg[h_x cos\alpha(t) + h_y sin\alpha(t)]$. However, because of the tension and slight stuttering effects on the imperfect bearing, $f(\alpha(t))$ might contain other unmodeled terms. In a stationary state, when $\ddot{\alpha}(t) = 0, \dot{\alpha}(t) = 0$ and $\dot{u}(t) = 0$, equation of motion (20) reduces to a time-invariant expression:

$$f(\alpha) = A\frac{L}{2}u. \quad (21)$$

Stationary $\alpha - u$ states can be closely resembled, when input signal $u$ is increased linearly over time with a very small rate of change while the measured $\alpha$ values are also captured during the experiment, as depicted in Figure 5. The connection between cohesive $\alpha - u$ value pairs can be well approximated by a fourth-order polynomial function $F(\alpha)$, which is considered as the static characteristic of the bi-rotor testbed:

$$u(\alpha) = F(\alpha) = 2\frac{L}{A}f(\alpha) = c_4\alpha^4 + c_3\alpha^3 + c_2\alpha^2 + c\alpha + c_0. \tag{22}$$

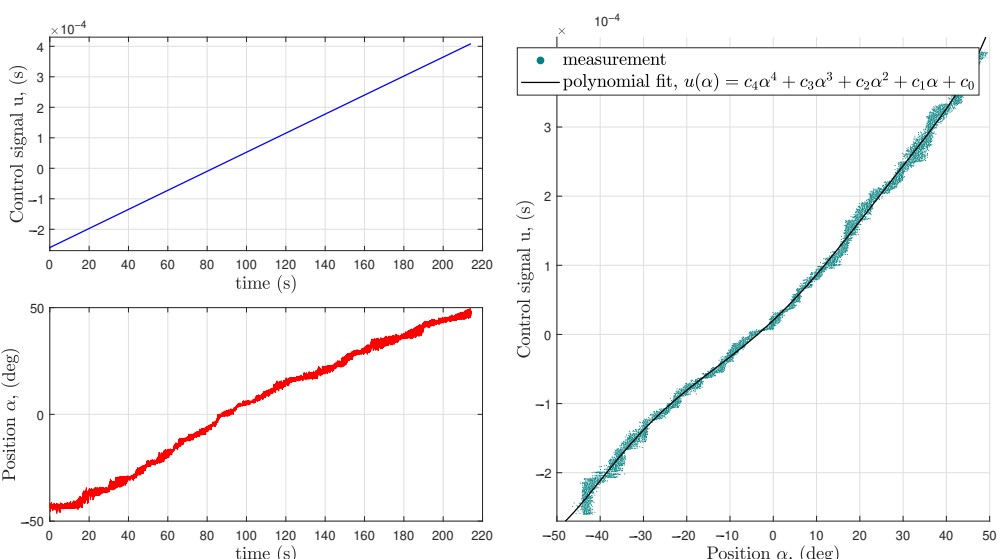

**Figure 5.** Results of the static measurement and the fitted curve for the static characteristic.

After the experimental determination of the static behavior, dynamic parameter values for $\Theta$ and $d$ are fine-tuned to match with the experimental step response results (Figure 6).

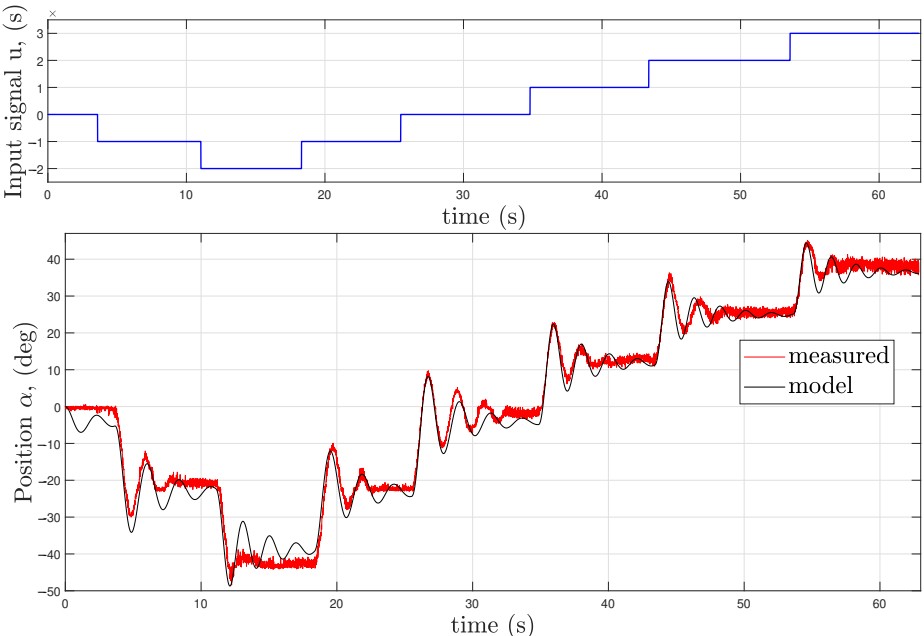

**Figure 6.** The output of the real and modeled plant.

The values of identified model parameters can be found in Table 2.

**Table 2.** Values, names, and units of identified model parameters.

| Parameter Name | Notation | Unit | Value |
|---|---|---|---|
| second order moment of inertia | $\Theta$ | [kg$\cdot$ m$^2$] | 0.0294 |
| viscous damping | $d$ | [N $\cdot$ m $\cdot$ (s/deg)] | $5.3 \times 10^{-7}$ |
| actuation time delay | $\tau$ | [s] | 0.08 |
| actuation rise time | $t_{rise}$ | [s] | 0.1578 |
| rotor axis distance | $L$ | [m] | 0.6 |
| arm mass (with battery) | $m$ | [kg] | 0.51 |
| input signal-force gain | $A$ | [N/s] | 2296.2 |
| static parameter | $c_0$ | [s] | $2.390 \times 10^{-5}$ |
| static parameter | $c_1$ | [s/deg] | $5.878 \times 10^{-6}$ |
| static parameter | $c_2$ | [s/deg$^2$] | $4.102 \times 10^{-8}$ |
| static parameter | $c_3$ | [s/deg$^3$] | $5.477 \times 10^{-10}$ |
| static parameter | $c_4$ | [s/deg$^4$] | $-1.119 \times 10^{-11}$ |
| simulation time step, cycle time for control | $\Delta t$ | [s] | 0.016 |

A prescribed control signal for a given desired acceleration $\ddot{\alpha}_{des}$ with current measured position and velocity values $\alpha, \dot{\alpha}$, based on (20) and (22), is calculated as:

$$u(t) = 2\frac{L}{A}[\Theta\ddot{\alpha}_{des}(t) + d\dot{\alpha}(t)] + 2\frac{L}{A}f(\alpha(t)) = 2\frac{L}{A}[\Theta\ddot{\alpha}_{des}(t) + d\dot{\alpha}(t)] + F(\alpha(t)). \quad (23)$$

Equation (23) serves as an inverse dynamic model for the controlled system. Here, it is worth mentioning that the inverse dynamic model described by the (23) neglects the actuation dynamics (time delay and rise time) since it is already compensated by the derivative effect of the kinematic block (see in the next section).

## 3. Results

The details of the realized CTC control scheme as it was first tested in Matlab Simulink is illustrated by Figure 7. For generating the rough nominal trajectory, the desired angular position was changed according to a predefined time schedule (0 degrees for 15 s, than +40, −40, +20, −20, +30, −30 and 0 degrees for 10–10 s). From these rough values, a smooth time-dependent path is generated by the smoothed trajectory generator, which consisted of four first-order serially connected low-pass filters with time constants of 0.2 s. This step is essential, since the second-order temporal derivative of the nominal position ($\ddot{\alpha}^N(t)$) has to be finite and two times continuously differentiable function which can be fed to the kinematic block (described by Equation (10)) to calculate the desired angular acceleration value $\ddot{\alpha}^{Des}(t)$.

In the first 5 s of the operation, control signal $u$ is set to zero resulting in $u_1 = u_2 = u_{lowest} = 1.3 \times 10^{-4}s$ signal levels for the left and right motors. This phase is used for spinning up the rotors to reach their lowermost effective actuation levels (see Figure 4). During this short "spinning up" phase, the arm executes small-amplitude free oscillation and the position is not controlled. Meanwhile, the measured angle is fed into the smooth trajectory planner, which thereby will be ready to generate a smooth transition path to the desired commanded position, when its output is already connected to the rough trajectory generator after 5 s. The measured transition between the uncontrolled first 5 s and the controlled state on the real physical system is also outlined later in the upper left graph in Figure 10.

During normal operation, the control signal $u(t)$ is calculated in two steps. First, the desired angular acceleration $\ddot{\alpha}^{Des}(t)$ is determined by the kinematic block in the function of nominal angular acceleration ($\ddot{\alpha}^N(t)$, originated from the smoothed trajectory generator), tracking error $e(t) = \alpha^N(t) - \alpha^{meas}(t)$, tracking error derivative $\dot{e}(t)$, and tracking error

integral $e_{int}$ according to Equation (10). Then, the desired acceleration $\ddot{\alpha}^{Des}(t)$, the measured current angular position $\alpha^{meas}(t)$, and its derivative $\dot{\alpha}^{meas}(t)$ are forwarded to the inverse dynamic model (Equation (23)) to obtain the current control signal $u(t)$, which here also plays the role of generalized control force $Q$ as also shown in Figure 1. To keep the controller output in a feasible range (between $u_{min} = 6 \times 10^{-4}$ s and $u_{max} = 6 \times 10^{-4}$ s, see again Figure 4), a saturation block is also added.

Besides the Simulink model, simulations were also carried out in the form of a Matlab script, where the simulation time step was $\Delta t = 0.016$ s and a simple first-order Euler integration scheme was used to get the angular velocity and position values. For imitating the sensor noise, a random number with normal distribution was added to the realized $\alpha(t)$ positions, and the standard deviation parameter $\sigma$ was 0.5053 degrees. The simulation was executed using six different $\Lambda$ trajectory tracking parameter values ($\Lambda = 1.75, 2.00, 2.25, 2.50, 2.75, 3.00$, the unit of $\Lambda$ is $s^{-1}$).

For performance evaluation of the proposed CTC scheme in the real world, a Circuit-Python code was generated and compiled on the microcontroller board belonging to the test-bed. The execution cycle time of the real-time CTC algorithm was $\Delta t = 0.016$ s (same value as the simulation time step). To attenuate tilt sensor noise without causing any additional time delay, fifty angle values were read from the tilt sensor in every computational cycle, and their mean value was utilized as the current $\alpha(t)$ value. Besides the mentioned straightforward "oversampling-averaging" technique, no other more advanced method was applied to improve the position signal quality. Instantaneous measured angular velocity was approximated by a simple numerical derivation ($\dot{\alpha} \approx \Delta\alpha/\Delta t$). Experimental measurements with a real-world test stand were also executed with six different $\Lambda$ values, lasting for 85 s.

Time series of the captured simulated and experimental data are compared in Figure 8. Good quality trajectory tracking was achieved in general both for simulated and experimental cases. The only appreciable difference between simulations and experiments is the behavior during the uncontrolled phase in the first five seconds, which can be explained by the fact that one of the rotors starts to spin slightly earlier (about 0.3–0.5 s) than the other, resulting in a short living non-zero torque acting on the beam. In the plots depicting control signal variations, upper and lower limits ($u_{min} = -6 \times 10^{-4}$ s, $u_{max} = 6 \times 10^{-4}$ s) are outlined by a dashed line. The CTC algorithm results in highly fluctuating control signals for all scenarios. However, the wobbling in $u(t)$ seems to be more intense in simulations, since the sensor noise level was slightly over-estimated. As the $\Lambda$ parameter increases, fluctuations become more and more significant, as well as $u(t)$ more likely tending to saturate at the limits of the usable signal range.

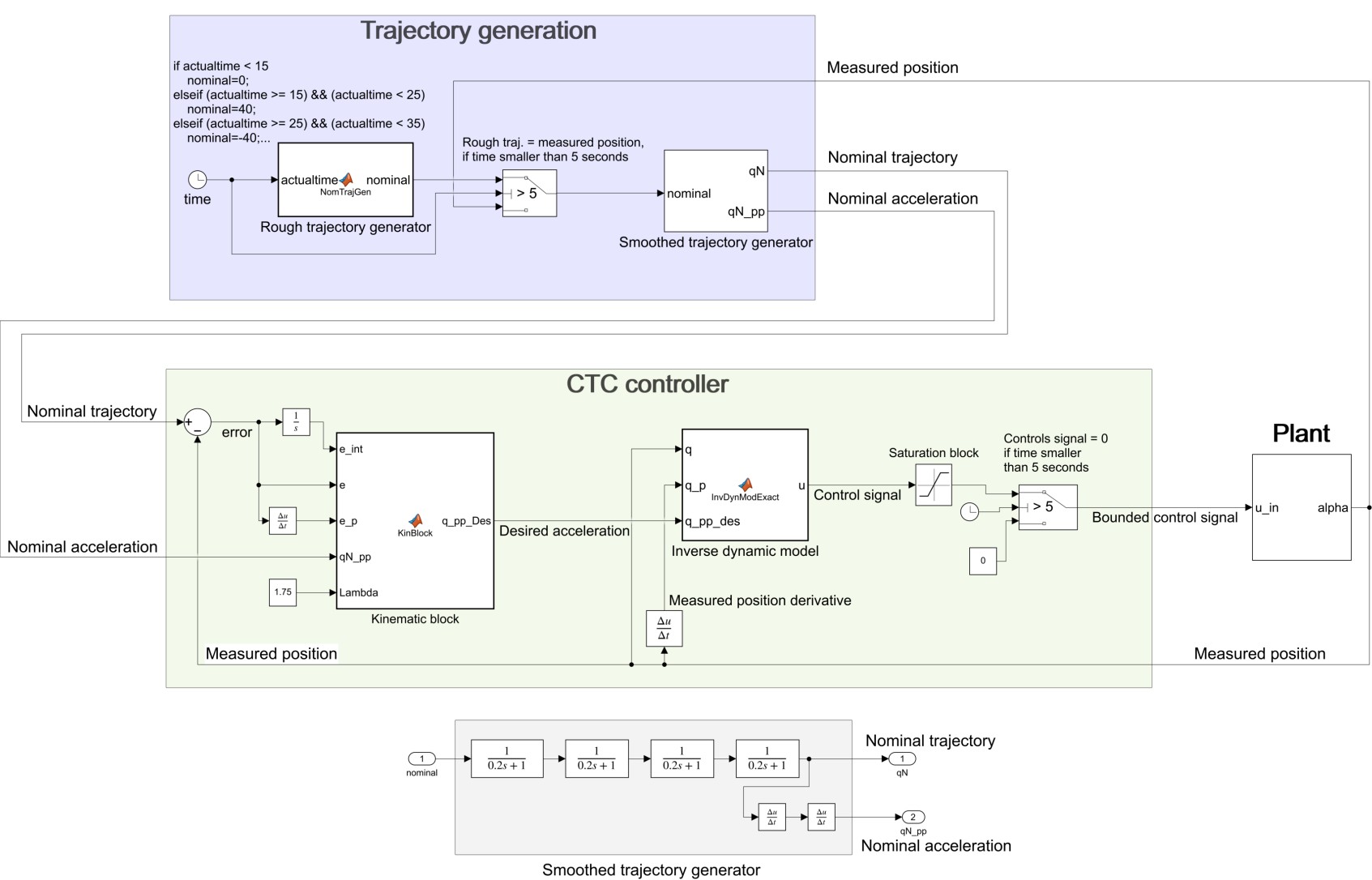

**Figure 7.** Details of the realized CTC control scheme modeled in Matlab Simulink.

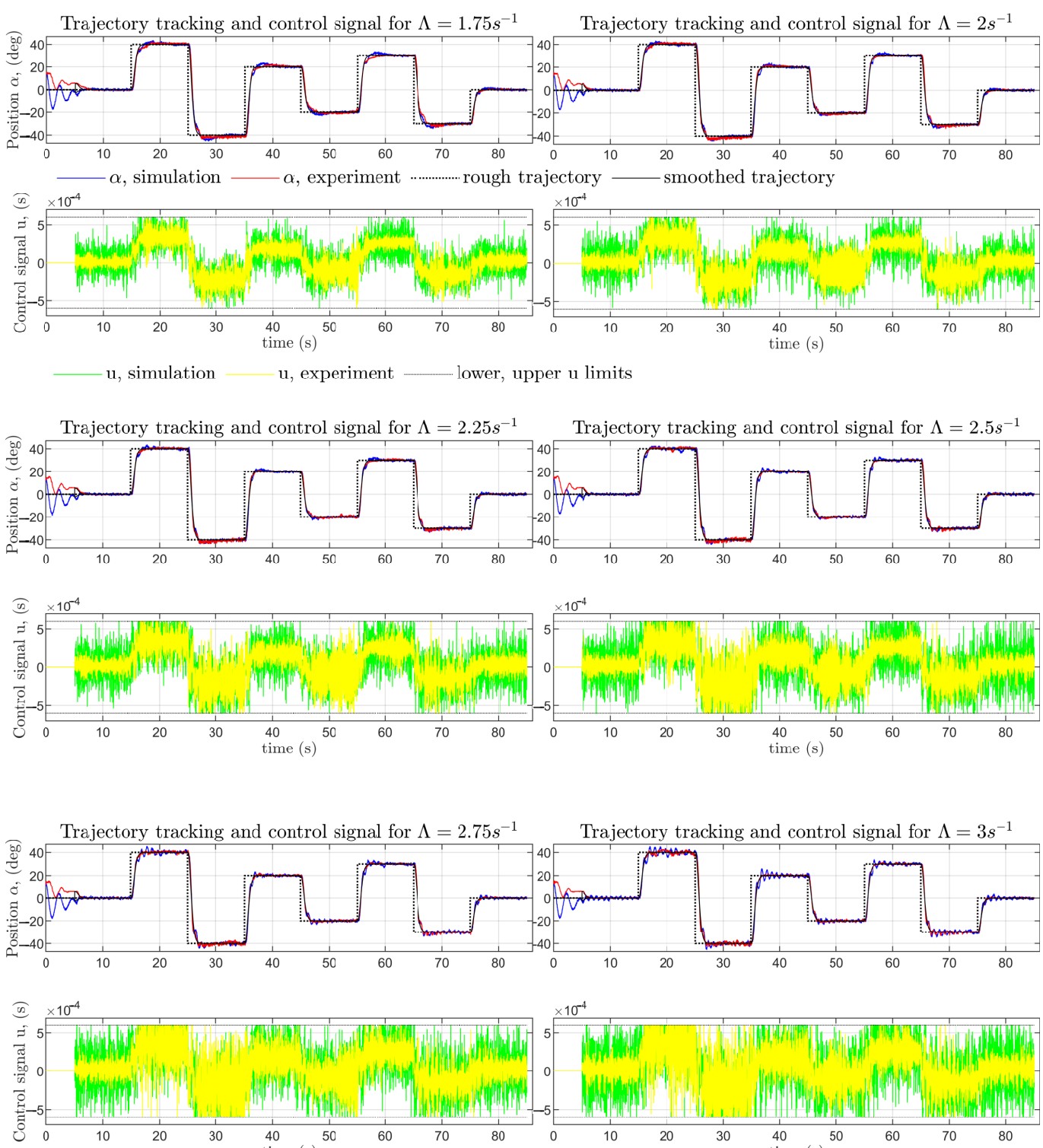

**Figure 8.** Experimental and simulation results for trajectory tracking with a "CTC" scheme in the case of six different $\Lambda$ values.

Small scale variations in the quality of trajectory tracking caused by changing parameter $\Lambda$ can be highlighted by evaluating the absolute integral error between the realized and nominal trajectory, while the control algorithm is active:

$$e_{int,abs,realized} = \int_{t=5s}^{t=85s} | \alpha^N(\xi) - \alpha^{realized}(\xi) | \, \mathrm{d}\xi. \tag{24}$$

Simulations and experimental measurements based on the aforementioned absolute integral error quantity are ranked in Figure 9.

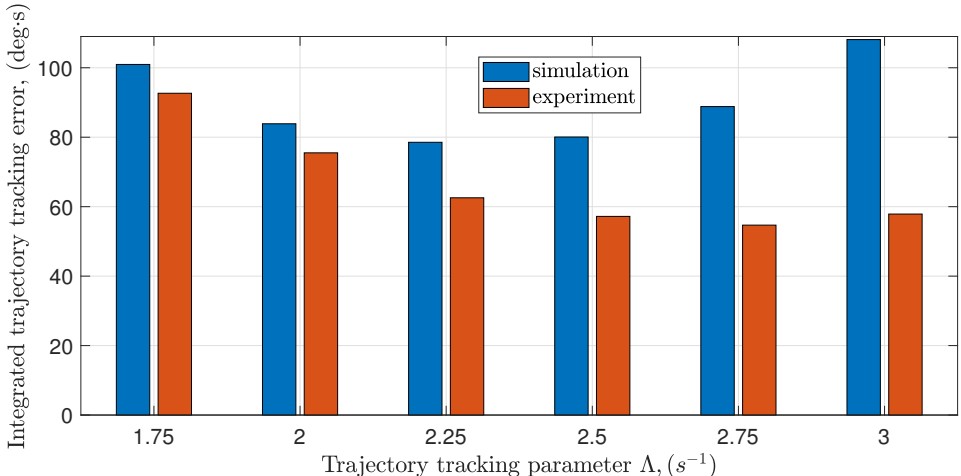

**Figure 9.** Absolute integral errors from simulations and measurements.

It can be observed that there is an optimal trajectory tracking parameter value that provides the smallest absolute integral error. The most advantageous value for $\Lambda$ is $2.75\,\mathrm{s}^{-1}$ in the case of experiments and $2.25\,\mathrm{s}^{-1}$ for simulations. This observation can be explained as follows: for smaller $\Lambda$ values, the trajectory tracking is less "aggressive", therefore some details of the nominal trajectory are not followed perfectly, while, for larger $\Lambda$ values, the controller is "overreacting", causing frequent saturation on the output, and, for this reason, some features of the desired path are again missed. Some detailed views of the best experimental trajectory followings with $\Lambda = 2.75\,\mathrm{s}^{-1}$ are shown in Figure 10.

To show that the proposed CTC scheme is not optimized just for a certain specific nominal trajectory, $\Lambda = 2.75\,s^{-1}$ experimental setting is also tested in the frequency domain by applying a sinusoidal wave as the desired path with a decreasing time period known as "chirp signal" (see Figure 11). Frequency is increased linearly over time from 0.001 Hz to 0.57 Hz, the peak-to-peak amplitude was 60 degrees with $-30$ minimal and $+30$ maximal values. The smooth transition between the controlled and uncontrolled phase is achieved by phase-shifting of the initial sine wave to match the last measured angle value at the end of the five seconds long "free oscillation" period. The trajectory tracking is almost perfect up to 0.3 Hz and a still acceptable quality path following can be observed up to 0.45 Hz (as a comparison: the natural frequency of the abandoned system, when oscillating freely is around 0.5 Hz). Low-grade path fidelity can be observed from 0.45 Hz to 0.65–0.7 Hz; stability loss is occurring around 0.75–0.8 Hz.

The change in the quality of trajectory tracking under the influence of external disturbance was first investigated by simulation (Figure 12). The duration of the applied external disturbance torque was 10 or 1 s, and the absolute value was one-third of the maximal achievable control torque (which is exerted when the lowest $u_{min}$ or the highest $u_{max}$ control signal values are sent permanently to the input signal distribution function depicted in Figure 4). When both short and long-duration disturbances occur or are removed, a transition period of about 10 s is needed for the algorithm to bring the controlled variable back to the vicinity of the prescribed nominal path.

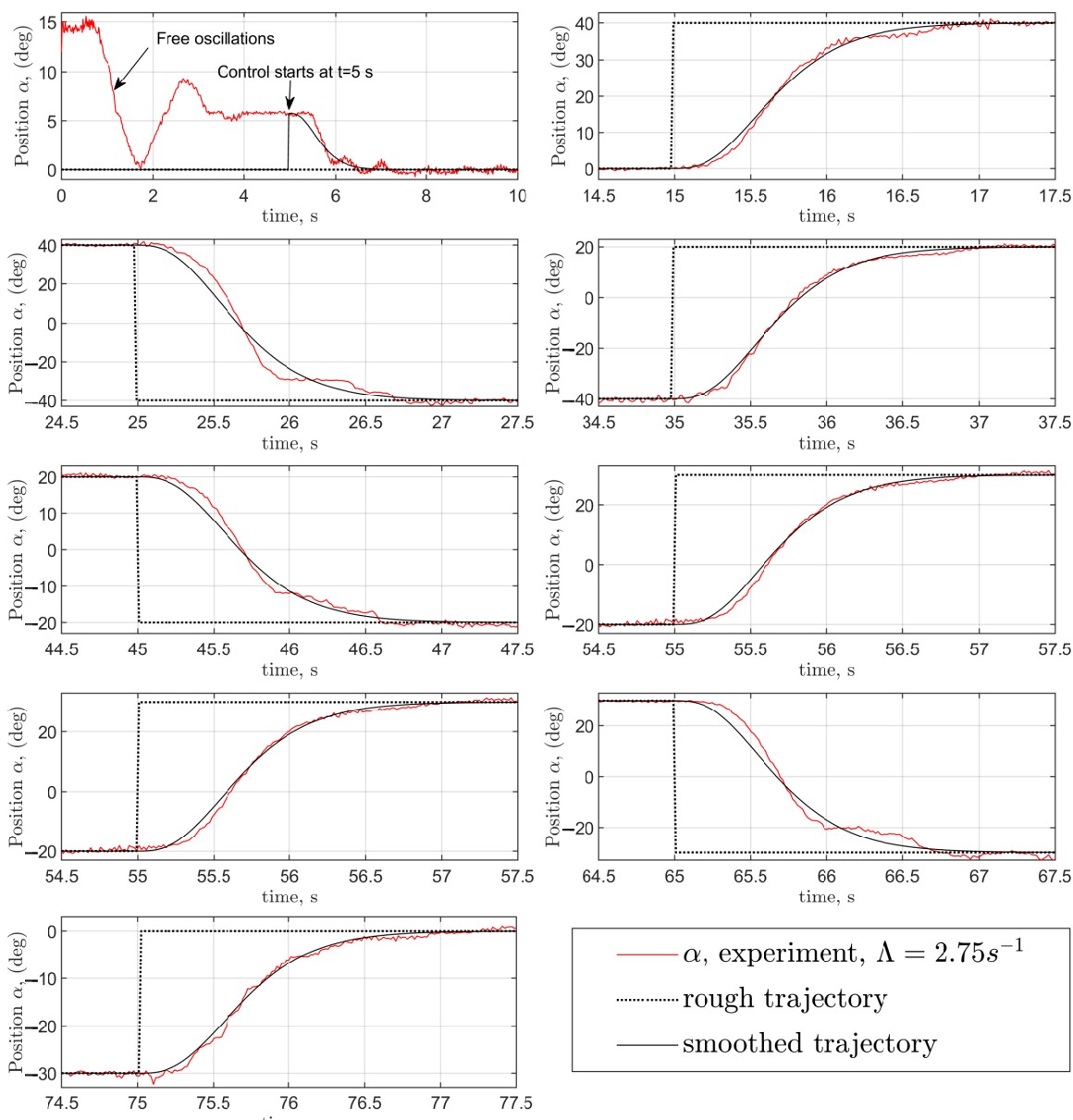

**Figure 10.** Detailed view of trajectory tracking from an experiment with $\Lambda = 2.75$ s$^{-1}$.

To show that the effect of external torque perturbations is also neutralized in real life, "qualitative" measurement series were elaborated, where the external disturbance was induced manually by pushing one of the arms belonging to the test stand arm (the measured results are shown in Figure 13, a similar magnitude of torque to the simulated disturbance could have been achieved by suspending weights to the arm, but the coordination of the weights' loading and unloading with the simulation load timing would have been too complicated). Each manual push lasted about 1 to 3 s. It is worth mentioning here that, in this experiment, the nominal trajectory was not predefined, but was generated in real-time by the microcontroller based on rough trajectory data provided by an incremental encoder knob. Each time, the experimenter turned the encoder knob by one position, the rough trajectory increased or decreased by 2 degrees, and this was fed to the smoothed trajectory generator .

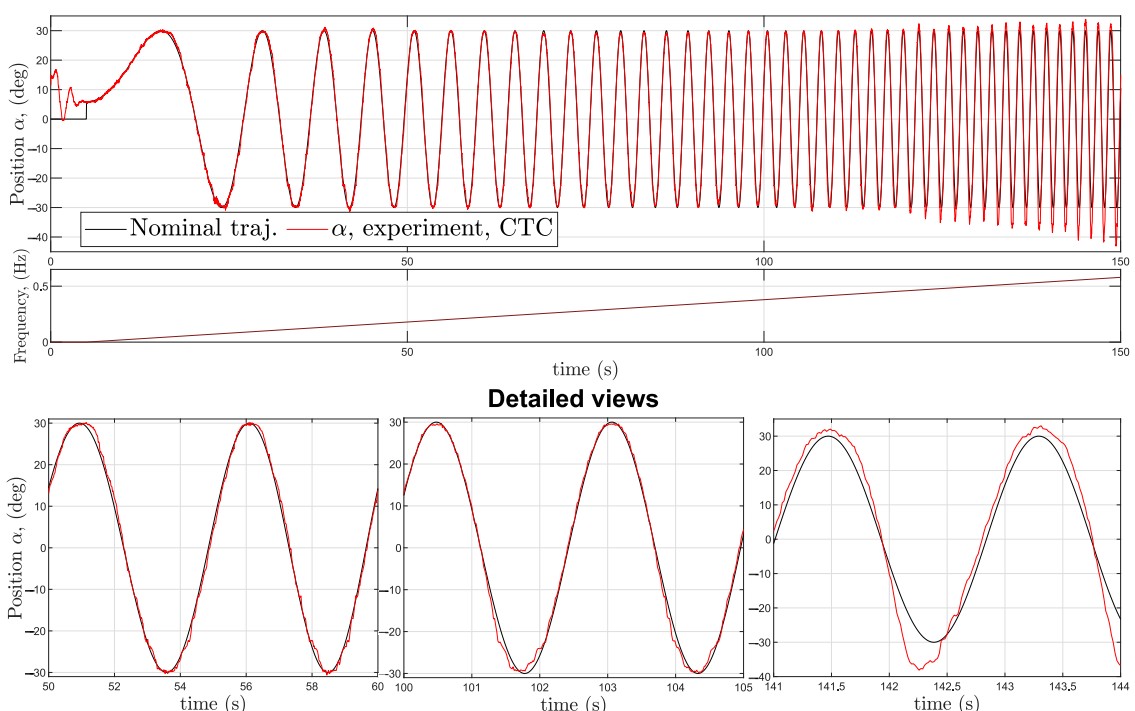

**Figure 11.** Experimental trajectory tracking for a sinusoidal trajectory with increasing frequency.

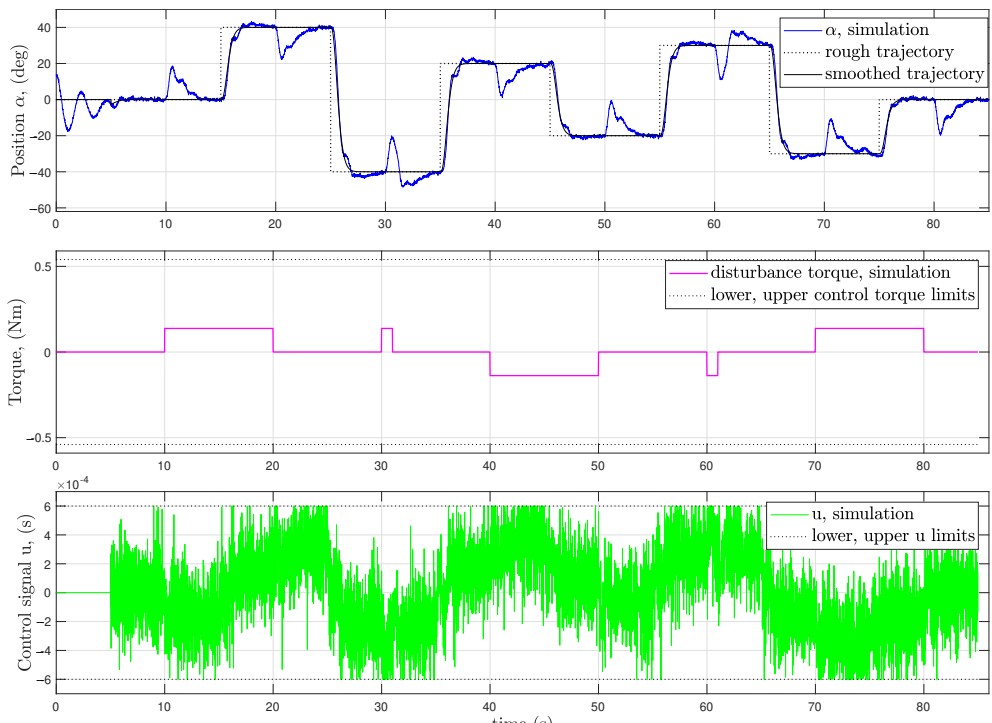

**Figure 12.** Trajectory tracking simulation in the presence of external torque disturbance, $\Lambda = 2.25$ s$^{-1}$. To illustrate the magnitude of the disturbance torque, the minimum and maximum control torque levels that can be applied by the rotors are shown as dashed lines on the middle graph.

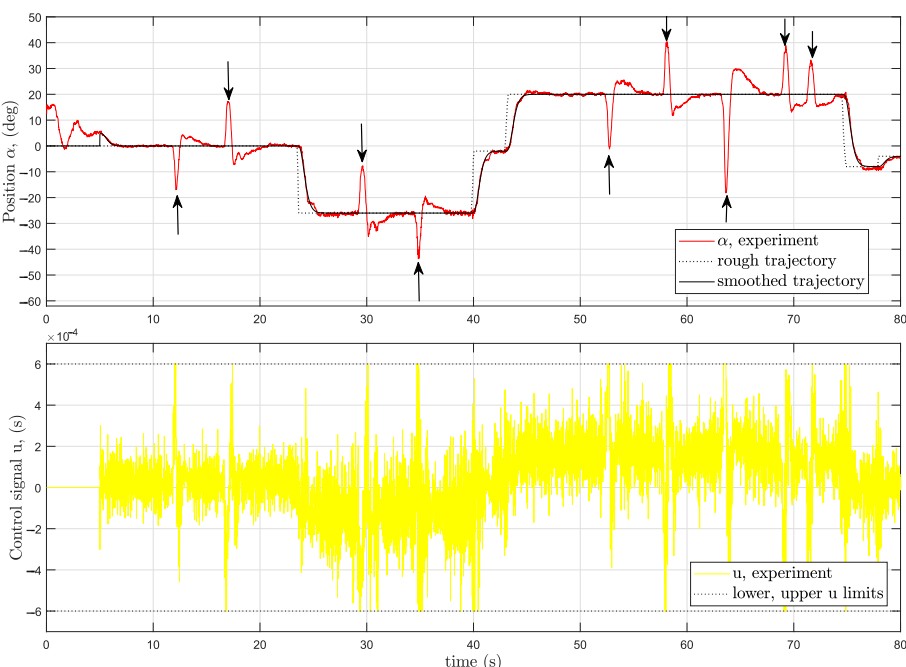

**Figure 13.** Experimental "qualitative" trajectory tracking measurement in the presence of external (manual) torque disturbance, $\Lambda = 2.75$ s$^{-1}$. Start time instances of manual disturbances are marked with arrows.

## 4. Performance Comparison of CTC and PID Compensator with a Nonlinear Feedforward Term

Since the desired nominal trajectory encompasses a wide range of motion (from –40 to +40 degrees), most conventional PID controller design methods, such as fitting a linear PID block to the linear approximated version of the original nonlinear system in a certain operating point, are out of the question. However, when the nonlinear part of the system model is supposed to be known or can be determined by carefully designed experimental measurements (e.g., approximation by polynomial functions, as was done previously in Section 2), PID compensation can be an option together with a nonlinear feedforward [4] term. In our particular case, the control signal for the bi-rotor testbed can be formulated as

$$u(t) = A\frac{L}{2}[F(\alpha(t))] + u_{PID}(t), \tag{25}$$

where

$$u_{PID}(t) = K_P[K_I e_{int}(t) + K_P e(t) + K_D \dot{e}(t))] \tag{26}$$

is a built-in by default PID controller block in Matlab Simulink (referred to as the "ideal form" of PID blocks), which can be "tuned" automatically to get a desired step response function. With controller in (25) (not considering the actuation time delay and rise time), the equation of motion (20) becomes:

$$\Theta\ddot{\alpha}(t) + d\dot{\alpha}(t) + f(\alpha(t)) = A\frac{L}{2}[F(\alpha(t))] + u_{PID}(t).) \tag{27}$$

Since $A\frac{L}{2}[F(\alpha(t))] = f(\alpha(t))$ (see Equation (22)), the troublesome nonlinear part is canceled out and the resultant model can be further treated as an LTI system. Deficiencies of the actuator dynamics (dead-time, rise time) can also be compensated up to a certain limit by appropriate $K_P, K_I, K_D$ choice. The Simulink model of the realized "PID with nonlinear feedforward" control scheme is depicted in Figure 14, where the feedforward linearizator block contains formula $A\frac{L}{2}[F(\alpha(t))]$ and the smoothed trajectory was generated in the same way as was done previously in the case of the CTC method.

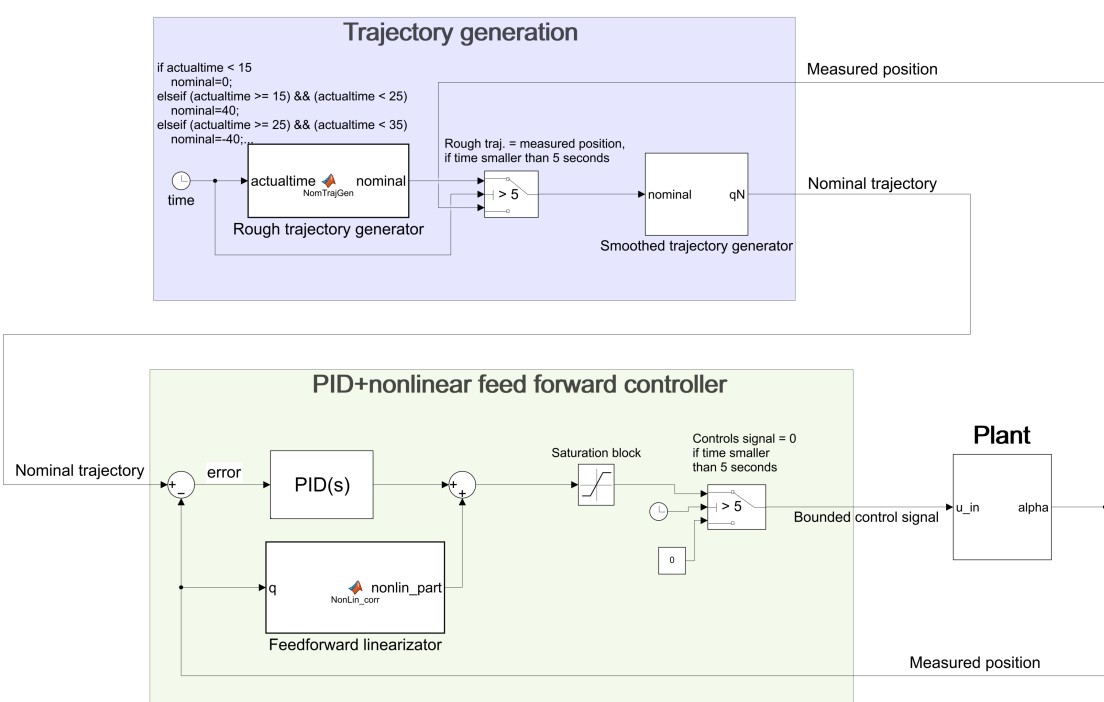

**Figure 14.** Details of the "PID with nonlinear feedforward" control scheme modeled in Matlab Simulink.

The obtained $K_P$, $K_I$, $K_D$ parameters from PID tuning are shown in Table 3. Since the lower computational demand is relative to the CTC algorithm, simulation time step and execution cycle time for the real-time controller board were halved from their original value. On the "D" channel, a first-order low-pass filter was inserted to attenuate the noise on the angular velocity signal (in addition, the aforementioned oversampling and averaging method for conditioning the measured angular position signal has also remained in use).

**Table 3.** Values of the tuned PID parameters used in "PID with nonlinear feedforward" control scheme.

| PID Parameter Name | Notation | Value |
|---|---|---|
| Proportional gain | $K_P$ | $5.33 \times 10^{-6}$ |
| Integral gain | $K_I$ | 1.50 |
| Derivative gain | $K_D$ | 0.37 |
| Derivative filter time constant in [s] | $T_D$ | 0.0079 |
| simulation time step, cycle time for control in [s] | $\Delta t$ | 0.008 |

Experimental and simulation results for trajectory tracking with "PID with nonlinear feedforward" control scheme are shown in Figure 15. Trajectory tracking is much less accurate relative to the CTC results, even with carefully tuned PID parameters. The constant sections in the desired path are reached with a considerable amount of over- or undershoots. These temporal peaks can be totally eliminated by choosing a less "aggressive" PID parameter set option, but, in this case, the settling time (the amount of time necessary to reach a constant value) expanded to 7–8 s, which generally resulted in even worse quality trajectory tracking. Experimental results for path following are again slightly better than the simulation ones, which can be explained by the "pessimistic" noise level approximation, and also by the sensitivity of the PID-based algorithm to the variations of the static and inertial parameters (see Figure 16 and the next two paragraphs about the model robustness).

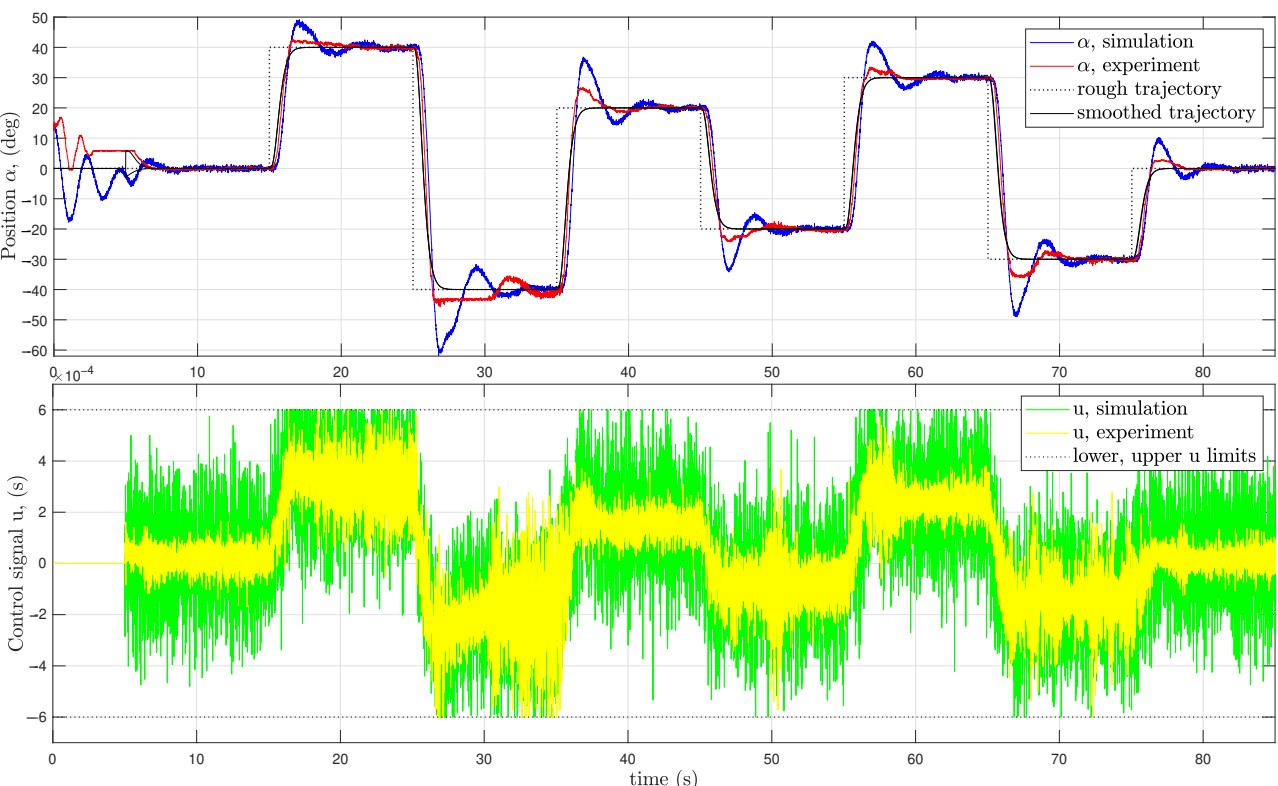

**Figure 15.** Experimental and simulation results for trajectory tracking with a "PID with nonlinear feedforward" control scheme.

To test the robustness and model parameter sensitivity of the two different proposed control schemes, a set of simulation studies were performed, where the original control scheme interacts with altered virtual plant versions (Figure 16, for CTC, $\Lambda = 2.25$ s$^{-1}$ case was chosen as the "original" controller since it had the smallest absolute integral error value). In the first set of simulations, only inertial parameter $\Theta$ was varied by $\pm 25\%$ relative to its identified value in Table 2; in the second set, only the viscous parameter $d$ was varied by $\pm 25\%$; in the third set, only time-delay $\tau$ was varied by $\pm 25\%$, and, finally, in the fourth set of simulations, all "static parameters" including $A, c_0, c_1, c_2, c_3, c_4$ were altered by $\pm 25\%$ at the same time in the plant model, while other system parameters were kept at a constant value.

While the CTC scheme was unaffected by all types of investigated model alternations, the "PID with nonlinear feedforward" algorithm shows a significantly different trajectory tracking behavior when the inertial or the static parameters were changed in the given range. The only disadvantage that can be mentioned in this comparison against the CTC is that it loses its stability when dead-time $\tau$ is increased drastically by factor 2.5 (0.2 s instead of the "true" 0.08 s), while the same time "PID with feedforward" still remains stable with very low quality trajectory tracking.

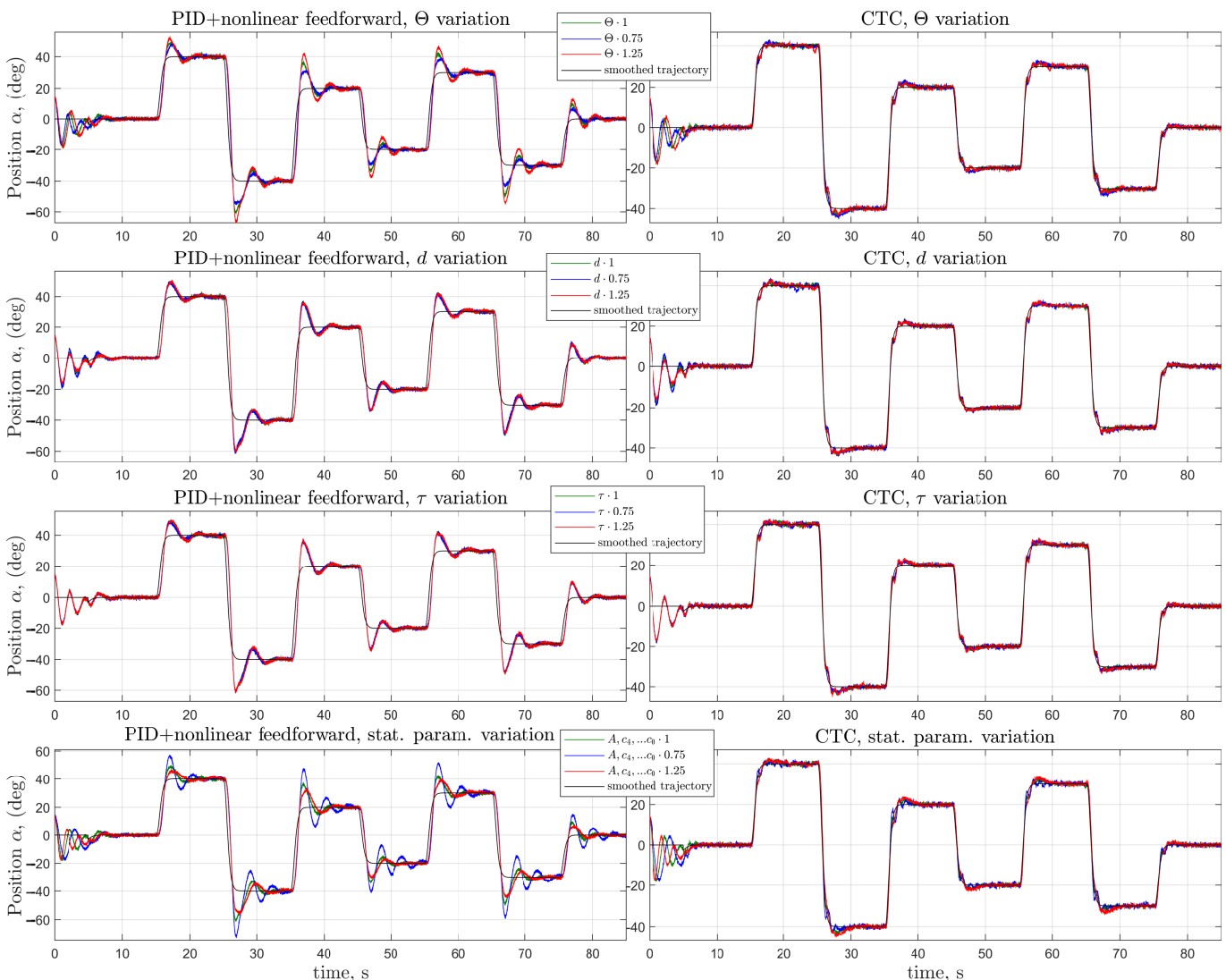

**Figure 16.** Simulation studies for investigating robustness and model parameter sensitivity; for CTC cases $\Lambda = 2.25 \text{ s}^{-1}$; for "PID with nonlinear feedforward" cases, the PID parameter set is from Table 3, $A, c_0, c_1, c_2, c_3, c_4$ values used in every case inside the CTC controller are from Table 2.

## 5. Conclusions

In highly nonlinear systems—e.g., in robotics, autonomous driving, navigation or attitude control of aerial vehicles—precise trajectory tracking is a crucial issue, especially if the system is loaded by time-delay as well. In this study, we have provided a generally applicable framework which aims to deal with this issue. We have developed and implemented an effective single parameter CTC control algorithm which was tested by various simulations and experiments that are ideal for applications where a specific nominal trajectory has to be followed precisely. The presented controller scheme preserves its stability and precision for a wide range of trajectory tracking parameter $\Lambda$ values even if the actuation dynamics are corrupted by severe time delays, saturation, significant external disturbances, and feedback sensor measurement noise. While designing the CTC controller, no high precision exact model is required, since it shows appreciable robustness against model parameter variations.

The robustness of the presented control scheme and its insensitivity to model parameter variations can be further improved by adding a fixed point iteration-based adaptive deformation block between the kinematic and inverse dynamic blocks in Figure 7, which modifies or "deforms" the desired acceleration based on the last measured realized acceler-

ation and the last calculated "deformed" acceleration value. Theoretical background and various simulation studies connecting to the CTC scheme supplemented by a fixed point transformation block can be found in [18–21]. It is worth noting that, while the theoretical background of this adaptive approach was elaborated in 1922 by Banach in his Fixed Point Theorem [22], the first proposal for its application in adaptive control was only published in 2009 in [21]. This method means a kind of "general framework" that can be filled in with various particular contents by specifying various functions, the use of which the task of finding the appropriate control signal can be transformed to iteratively finding the fixed point of a function. The function called "Robust Fixed Point Transformation" was the first example that was published in [21]. Later different functions were suggested and investigated via simulations in [23–25] that mean potential solutions. In the near future, experimental performance investigation of fixed point iteration based methods will hopefully be carried out by the help of the presented twin rotor test platform.

**Author Contributions:** Conceptualization, J.K.T. and G.E.; methodology, Á.V.; software, Á.V.; validation, Á.V.; formal analysis, J.K.T.; investigation, Á.V.; resources, I.R. and G.E.; data curation, Á.V.; writing—original draft preparation, Á.V. and J.K.T.; writing—review and editing, Á.V. and J.K.T.; visualization, Á.V.; supervision, J.K.T. and G.E.; project administration, G.E.; funding acquisition, I.R. and G.E. All authors have read and agreed to the published version of the manuscript.

**Funding:** G.E. was supported by the Eötvös Loránd Research Network Secretariat under Grant No. ELKH KÖ-40/2020 (Development of cyber-medical systems based on AI and hybrid cloud methods). Project No. 2019-1.3.1-KK-2019-00007 has been implemented with the support provided from the National Research, Development and Innovation Fund of Hungary, financed under the 2019-1.3.1-KK funding scheme.

**Data Availability Statement:** The Matlab and Python codes, Simulink models, measurement data, video links are available at: https://drive.google.com/drive/folders/1hNdgYBwFCSzv84MubIOa rzTUEdwEKY2g?usp=sharing.

**Acknowledgments:** The research was supported by the Doctoral School of Applied Informatics and Applied Mathematics at Óbuda University.

**Conflicts of Interest:** The authors declare no conflict of interest.

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
