# Peer review of "Experimental and Simulation-Based Performance Analysis of a Computed Torque Control (CTC) Method Running on a Double Rotor Aeromechanical Testbed"

_electronics, doi:10.3390/electronics10141745_

Round 1
Reviewer 1 Report
The paper deals with a new modification of the so-called Computed Torque Control (CTC) algorithm, which was experimentally tested and evaluated on a double rotor system. An extensive analysis of the obtained experiments was performed.
The main contribution of the paper can be seen in using a CTC algorithm, which requires to adjust only one parameter in contrast to other control algorithms. Therefore, the topic of the paper can be seen as beneficial for the control theory. Further, the paper is well structured and the experimental part contains suitable summary of achieved results, too.
I have some remarks for improving the quality of the paper:
- The paper needs still a careful checking and correcting the grammar, mainly mixtures of singulars and plurals in some sentences. In English, only the terms ‘control’ and ‘controllers’ should be used in the given topic area and not ‘regulator’ or ‘regulate’ as used sometimes.
- In most figures too small fonts are used.
- Too old and too many literature sources are used. The historical introduction into the topic is not necessary.
- Please add a block diagram of the proposed research methodology.
Author Response
The paper deals with a new modification of the so-called Computed Torque Control (CTC) algorithm, which was experimentally tested and evaluated on a double rotor system. An extensive analysis of the obtained experiments was performed.
The main contribution of the paper can be seen in using a CTC algorithm, which requires to adjust only one parameter in contrast to other control algorithms. Therefore, the topic of the paper can be seen as beneficial for the control theory. Further, the paper is well structured and the experimental part contains suitable summary of achieved results, too.
I have some remarks for improving the quality of the paper:
Point 1: The paper needs still a careful checking and correcting the grammar, mainly mixtures of singulars and plurals in some sentences. In English, only the terms ‘control’ and ‘controllers’ should be used in the given topic area and not ‘regulator’ or ‘regulate’ as used sometimes.
Response 1: The manuscript has been thoroughly checked to correct the remaining grammar and spelling mistakes, the altered text sections are marked with red in the manuscript PDF.
The term 'regulator' has been used for stylistic reasons to avoid the large and frequent repetition of the terms 'control' and 'controller'. However, as it can be misleading (in many places the term 'regulator' is indeed used for open-loop control solutions), we have replaced the term 'regulator' with 'controller' everywhere, as requested by the reviewer.
Point 2: In most figures too small fonts are used.
Response 2: Axis and plot legends are enlarged in Figures 5-6, 8-13, 15-16. In the case of Figure 7 (Details of the realized CTC control scheme modeled in Matlab Simulink), there is no simple way to increase font size without destroying the figure integrity, therefore, the whole illustration is rotated by 90 degrees and enlarged significantly. A similar solution is applied for Figure 14 (Details of the "PID with nonlinear feedforward" control scheme modeled in Matlab Simulink), without the rotation.
Point 3: Too old and too many literature sources are used. The historical introduction into the topic is not necessary.
Response 3: The unnecessary historical introduction (between rows 21-62 in the first version of the manuscript) into the topic has been deleted, the first section directly starts with the introduction of the CTC control. Therefore, about 10-15 old literature sources are deleted, also.
Point 4: Please add a block diagram of the proposed research methodology
Response 4: A new block diagram has been added to the manuscript (Figure 2: Block diagram of the methodology used in the article), which summarizes the research methodology, the carried out investigations and also serves as a graphic table of contents.

Reviewer 2 Report
Dear authors,
You did an interesting work but the presentation of conclusion does not satisfy. In the conclusion should briefly summarize the findings and highlight the original part of the work presented in this manuscript. There, please modify it.
Author Response
Response to Reviewer 2 Comments
Point 1: You did an interesting work but the presentation of conclusion does not satisfy. In the conclusion should briefly summarize the findings and highlight the original part of the work presented in this manuscript. There, please modify it.
Response 1: We have slightly reformulated the Conclusion section to highlight the original part of the work presented in this manuscript (highlighted with red in the manuscript PDF). We have also added a new figure (Figure 2: Block diagram of the methodology used in the article.) at the beginning of the manuscript, which graphically summarizing all the investigations executed in the framework of the article
